# Strength and Fracture Toughness of Hardox-400 Steel

**Ihor Dzioba \*** and **Robert Pała**

Department of Machine Design, Faculty of Mechatronics and Mechanical Engineering,
Kielce University of Technology, Al. 1000-lecia PP 7, 25-314 Kielce, Poland; rpala@tu.kielce.pl
\* Correspondence: pkmid@tu.kielce.pl; Tel.: +48-41-3424-303

**Abstract:** This paper presents results of strength and fracture toughness properties of low-carbon high-strength Hardox-400 steel. Experimental tests were carried out for specimens of different thickness at wide temperature range from −100 to 20 °C. The dependences of the characteristic of material strength and fracture toughness on temperature based on experimental data are shown. Numerical calculation of the stress and strain distributions in area before crack tip using the finite element method (FEM) was done. Based on results of numerical calculation and observation of the fracture surfaces by scanning electron microscope (SEM), the critical local stress level at which brittle fracture takes place was assessed. Consideration of the levels of stress and strain in the analysis of the metal state at the tip of the crack allowed to justify the occurrence of the brittle-to-ductile fracture mechanism. On the basis of the results of stretch zone width measurements and stress components, the values of fracture toughness at the moment of crack initiation were calculated.

**Keywords:** Hardox-400 steel; strength properties; fracture toughness; critical local stress level

## 1. Introduction

During recent years, new low-carbon ferritic steels have been designed and produced which are characterized by their high yield stress values or high hardness with maintaining a good level of plasticity. Development of this type of steel and carrying out research to determine their mechanical properties is of great importance for their applications in various industries, including building construction, automotive, aerospace, and others.

To achieve such a high strength, these steels are subjected to thermomechanical processing (TMP) [1–5]. In these works, an accurate analysis of the microstructure, resulting from slightly different TMPs and various element contents, is also presented. Generally, it is a multiphase microstructure that consists of ferrite, bainite, retained austenite, and martensite. Changes in the microstructure lead to changes in material properties. The most frequently studied were the effect of microstructural changes on strength characteristics, plasticity and hardness [6–12], and less frequently—on impact resistance [13].

In a few publications, the results of fracture toughness test of high-strength steels (HSS) have been presented [14–21]. Lack of accurate information on the fracture toughness of HSS does not allow to properly assess the strength of structural elements made of these steels and assess the parameters of their safe use [22]. Therefore, this publication presents the results of investigations of mechanical properties of Hardox-400 steels in the temperature range of their use.

During production, the TMP technology aims to achieve uniform yield strengths (e.g., $\sigma_y \geq 960$ MPa for S960QC) or hardness (e.g., HV $\geq 400$ for Hardox) for each plate thickness. As a result, negligible differences in the microstructure for different plate thicknesses and across plate thickness can be observed. These microstructural differences lead to small changes in strength properties across thickness, but fracture toughness values change considerably.

The influence of temperature in wide range from −100 to 20 °C on strength and fracture toughness characteristics were tested. Special attention allowed to determine fracture toughness characteristics. The change of critical value of fracture toughness with specimens' thickness was tested. Stress and strain distributions in front of the cracks were calculated by numerical modeling and FEM. Based on the results of numerical calculations, the values of critical local stress were calculated, the achievements of which enables a realization of brittle fracture in the specimens.

This paper presents a review of the results which were obtained in testing of high-strength steels: S960QC and Hardox-400 since 2009. Several papers published in this time period have reported the results obtained during investigation of these steels [16–21]. However, during next testing, the results were complemented and more exact. This paper includes an accurate data received within all test time. This paper mainly presents the results obtained during testing high-strength Hardox-400 steel. The results for strength and fracture toughness properties of the S960QC steel are qualitatively similar.

## 2. Materials and Methods

### 2.1. Microstructure and Hardness of Tested Plate of Hardox-400 Steel

The plates of Hardox-400 steel of 30 mm thickness was produced using a controlled thermomechanical treatment. The chemical composition of Hardox-400 steel is shown in Table 1. The microstructure of this steel is tempered martensite–bainite (Figure 1a). The grain sizes were within the range of 5–20 μm. A separate isolated large nonmetallic inclusions of oxides and sulphides (Figure 1b) and particles of titanium nitrides enriched with Nb (Figure 1c) of size 1.0–2.5 μm were present in this steel. Qualitative identification of the chemical composition of these particles was carried out by energy dispersive X-ray spectroscopy analysis (EDX) (JEOL, Peabody, MA, USA), (Figure 2). Additionally, numerous particles of carbides precipitates of sizes 50–300 nm (Figure 1d) were observed in areas of ferrite. The difference of microstructure between layers in the middle part of a plate and near surface is insignificant.

**Table 1.** The chemical composition of Hardox-400 steel according to [23].

| C, % | Si, % | Mn, % | P, % | S, % | Cr, % | Ni, % | Mo, % | B, % |
|------|-------|-------|------|------|-------|-------|-------|------|
| 0.32 | 0.70  | 1.60  | 0.025| 0.01 | 2.40  | 1.50  | 0.60  | 0.004|

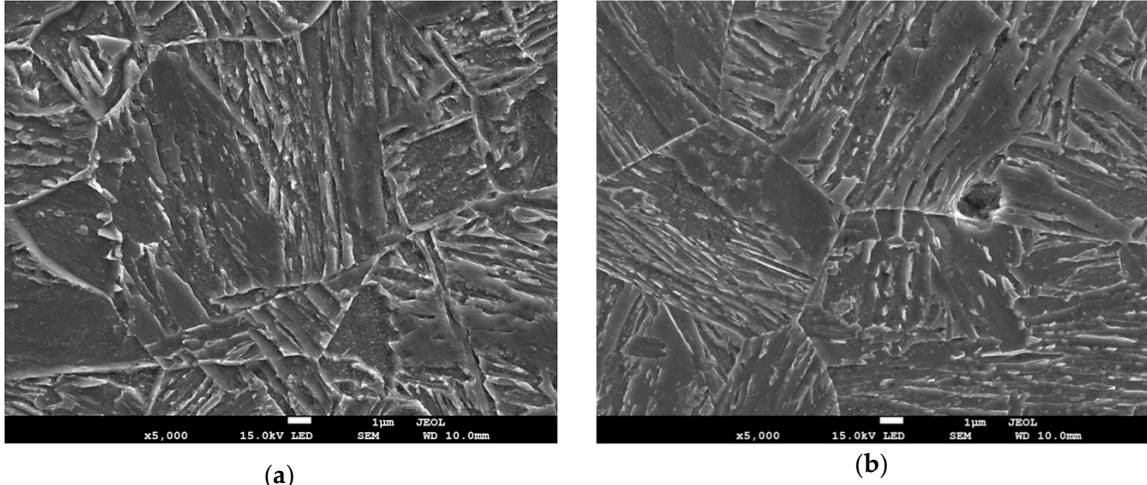

(a)      (b)

**Figure 1.** *Cont.*

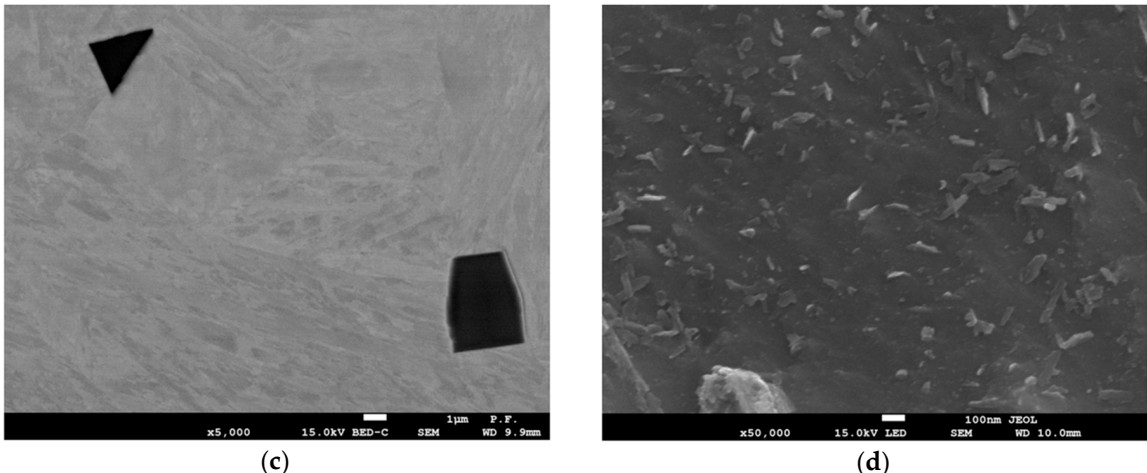

**Figure 1.** (**a**) Microstructure of Hardox-400 steel (**b**,**c**) with inclusions and (**d**) particles of carbides precipitates in ferritic regions.

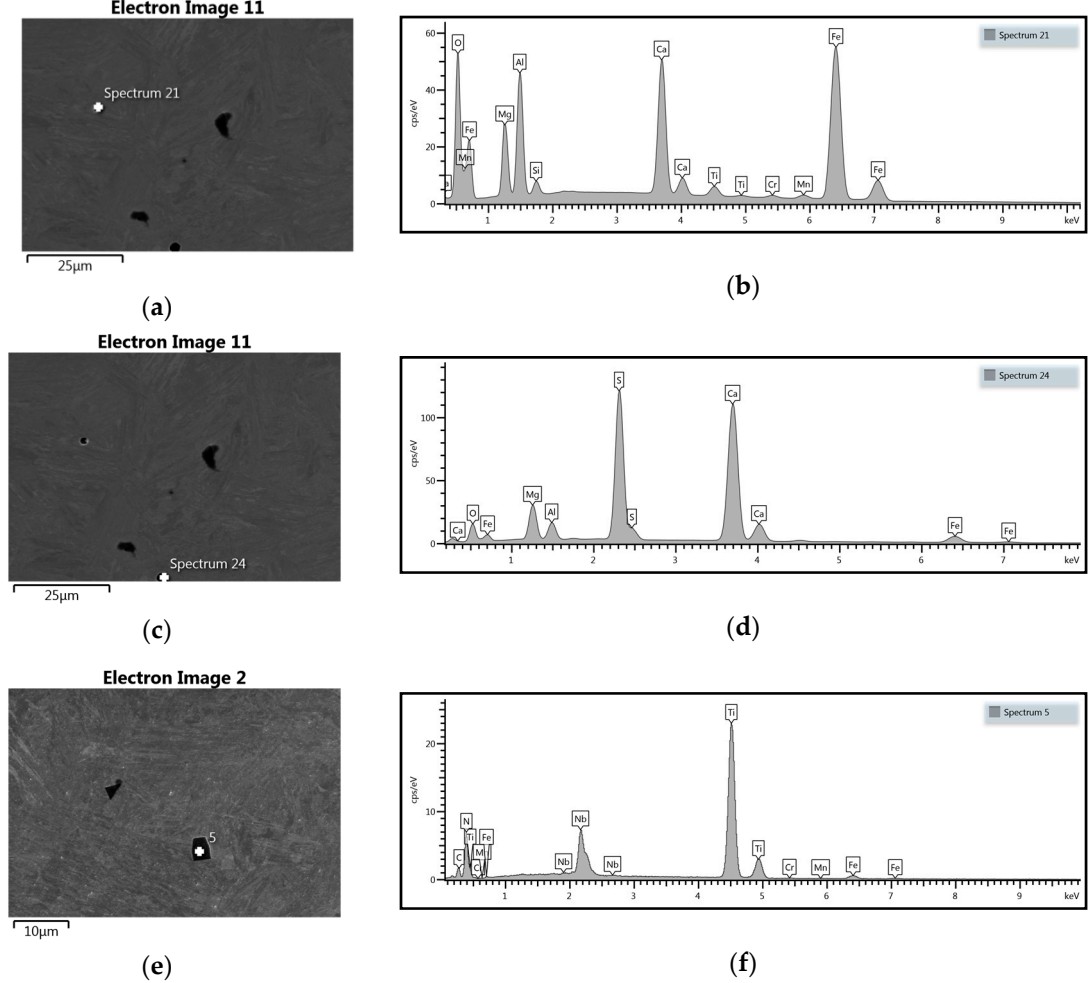

**Figure 2.** Large particles of inclusions and results of the energy dispersive X-ray spectroscopy (EDX) analysis of (**a**,**b**) oxides; (**c**,**d**) sulfides; (**e**,**f**) titanium nitrides enriched with Nb.

As a result of manufacturing by thermomechanical treatment, the hardness distributions changed along the plate thickness (see Figure 3a). In the middle part of a plate of 30 mm thick, the average level of hardness was about 350 HV10, but the near surface of a plate is about 400 HV10, and on a surface of

a plate it exceeds 425 HV10. The significant and considerable scatter of hardness data along thickness was observed.

Tests were conducted on specimens that were cut out from a middle part of a plate of Hardox-400 steel, where hardness level was lower (Figure 3a,b). For determination of strength properties, cylindrical specimens were used with diameter of 5 mm and $L_0$ = 25 mm that were tested at tensile loading. Fracture toughness characteristics were obtained using Single Edge Notch Bend Specimens (SENB) ($B$ = 1.0–24.0 mm; $W$ = 24 mm; $S$ = 96 mm; $a_0/W$ = 0.5). Due to the frequent utilization of these types of steel in constructions in regions with low temperature, both strength and fracture toughness properties were determined in a wide temperature range: from −100 to 20 °C.

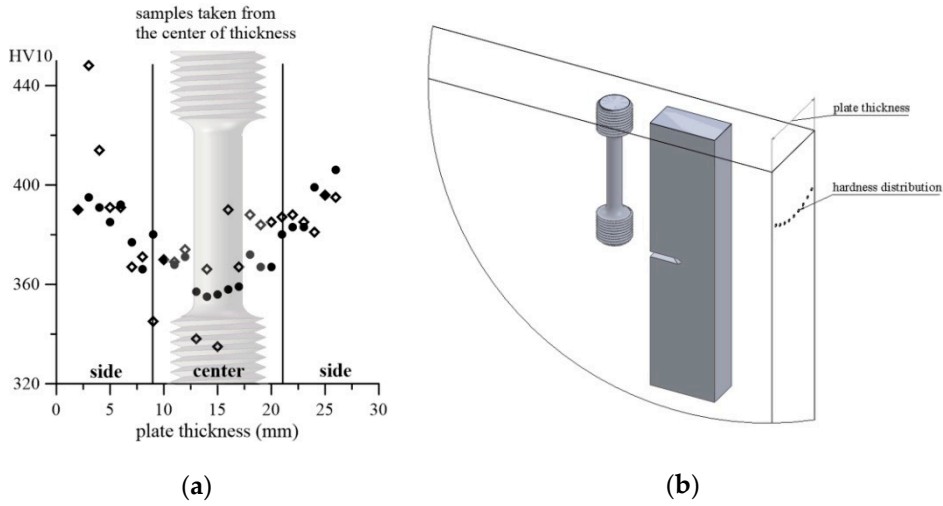

(**a**)         (**b**)

**Figure 3.** (**a**) Hardness distribution through the plate thickness; (**b**) the scheme of cutting out a specimen.

All experimental uniaxial tensile and fracture toughness tests were carried out using MTS (MTS Systems Corporation, Eden Prairie, MN, USA)and Zwick (Zwick Roell Group, Ulm, Germany) universal testing machine with system of automatic control of operating and data recording. The signals of load cell and specimen elongation extensometer were recorded during uniaxial tensile tests, and during fracture toughness tests the signals of load cell, extensometers of displacement of specimen in loading point and crack mouth opening, were recorded in real time and then used for material properties determination. The thermal chamber and vapor of liquid nitrogen was used for specimens tested at lowering temperatures. Temperature measurement made by thermocouple, which was placed immediately on the tested specimen, with accuracy of ±0.1 °C. All metallographic, fractographic tests and analyses were carried out by the scanning electron microscope (SEM) JSM-7100F (JEOL, Peabody, MA, USA).

*2.2. Temperature Influence on Strength Properties of Hardox-400 Steel*

The strength properties through the thickness were not uniform, but differences of these properties obtained on specimens cut from the middle layer and near surface were not large. The average values of strength characteristics for specimens from middle part of a plate were: $\sigma_y$ = 950 MPa, $\sigma_{uts}$ = 1205 MPa; and for specimens from materials near surface: $\sigma_y$ = 975 MPa, $\sigma_{uts}$ = 1220 MPa. However, during determining the strength properties of Hardox-400 steel, a wide scatter of data was observed for both localizations of specimens from the middle part and near the surface of plates. In the following part of the paper, we present the results according to the specimens cut out from the central layer of plates.

Based on the data recorded during uniaxial tensile tests, the nominal and real strength characteristics of Hardox-400 steel were received. Figure 4 shows the stress–strain ($\sigma$–$\varepsilon$) graphs for different test temperatures (−100–20 °C). While on scatter of data, the values of strength and plastic characteristics of Hardox-400 steel are increased with the decrease of temperature. These trends are

well described by linear functions. The average values of the characteristics for test temperature are presented in the Table 2. There are also the equations obtained by fitting of linear function of all points received in all tested temperatures. The data of the true $\sigma-\varepsilon$ graphs are necessary to create material model for the numerical calculation, which will be presented in the next part of the paper.

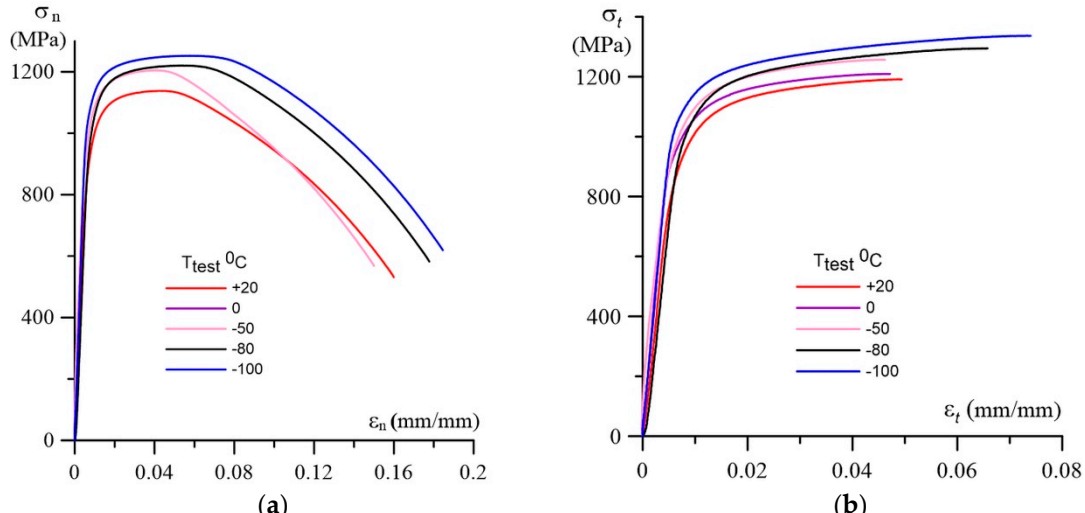

**Figure 4.** The diagrams of $\sigma-\varepsilon$ for the Hardox-400 steel: (**a**) Nominal and (**b**) true.

**Table 2.** The strength and plasticity properties of Hardox-400 steel.

| Hardox-400 | 20 °C | | 0 °C | | −50 °C | | −80 °C | | −100 °C | | Equation |
|---|---|---|---|---|---|---|---|---|---|---|---|
| | Value | scat. % | Value | scat. % | Value | scat. % | Value | scat. % | Value | scat. % | |
| $\sigma_{y\_n}$; $\sigma_{y\_t}$ (MPa) | 947 955 | 1.1 | 958 966 | 0.4 | 990 999 | 0.5 | 1020 1030 | 0.1 | 1053 1062 | 1.4 | $\sigma_{y\_t} = -0.8534 \cdot T_{test} + 961.91$ |
| $\sigma_{uts\_n}$; $\sigma_{uts\_t}$ (MPa) | 1149 1205 | 1.1 | 1162 1212 | 0.2 | 1195 1254 | 0.7 | 1221 1297 | 0.3 | 1245 1327 | 1.1 | $\sigma_{uts\_t} = -1.0138 \cdot T_{test} + 1212.39$ |
| $E\_n$; $E\_t$ (GPa) | 180 181 | 2.1 | 179 180 | 0.8 | 176 177 | 3.1 | 173 174 | 5.9 | 202 203 | 8.3 | $E\_t = -0.0767 \cdot T_{test} + 178.37$ |
| $n\_n$; $n\_t$ | 10.9 9.8 | 4.6 | 10.2 9.3 | 2.8 | 10.96 9.76 | 4.7 | 12.42 10.51 | 1.0 | 14.36 11.72 | 7.3 | $n\_t = -0.0131 \cdot T_{test} + 9.57$ |
| $\varepsilon_{uts\_t}$ | 0.05 | 6.4 | 0.045 | 9.0 | 0.052 | 13.7 | 0.066 | 1.1 | 0.071 | 2.7 | $\varepsilon_{uts\_t} = -0.0002 \cdot T_{test} + 0.05$ |
| $\varepsilon_{c\_n}$ | 0.16 | 5.7 | 0.134 | 14.5 | 0.162 | 4.2 | 0.175 | 1.7 | 0.184 | 0.2 | $\varepsilon_{cn} = -0.0003 \cdot T_{test} + 0.15$ |

*2.3. The Influence of Temperature and Specimen Thickness on Fracture Toughness Characteristics*

Fracture toughness characteristics were determined according to the recommendations of ASTM standards [24,25]. A drop potential and compliance change methods were used to obtain the critical values of *J*-integral, $J_C$, when ductile toughness at crack initiation and propagation took place. While, if crack initiation and propagation occurred as brittle, the critical value of *J*-integral, $J_C$, was calculated according to the formula: $J_C = \eta E_C / (B(W - a_0))$. If all requirements of ASTM standards according to specimen dimensions and procedures were satisfactory, the $J_C$ are the material characteristic and denoted as $J_{IC}$. In the research program, SENB specimens with different thickness were tested. For some of them, the requirement on critical thickness was not observed, so the critical value of *J*-integral was not classified as $J_{IC}$, but as $J_C$. All results of the critical fracture toughness values are presented in *J*-integral units; for recalculation to stress intensity factor (SIF) units, the equation $K_C = (E J_C / (1 - \nu^2))^{0.5}$ should be used.

A significant scatter range was typical for fracture toughness critical values $J_C$ of Hardox-400 steel. A scatter range increased when fracture mechanisms of subcritical crack growth in SENB specimens are fully ductile, or firstly ductile and then brittle. When subcritical crack growth in specimens occurred

according to the pure brittle-by-cleavage mechanism, the scatter of fracture toughness data was small. An example of data distributions of $J_C$ of test temperature $T$ for specimens of 2 mm and 12 mm are shown in Figure 5. The large data scatter was observed in the regions of ductile fracture for specimens of thickness 2 mm (see Figure 5a) and in the region of ductile-to-brittle change for specimens of thickness 2 mm and 12 mm (see Figure 5a,b). In specimens of both thicknesses in brittle fracture regions, the scatter of values were slight (see Figure 5a,b). Qualitatively, similar types of distributions results were observed for specimens of thickness 1.0, 2.0 mm, and from 8.0 to 24.0 mm.

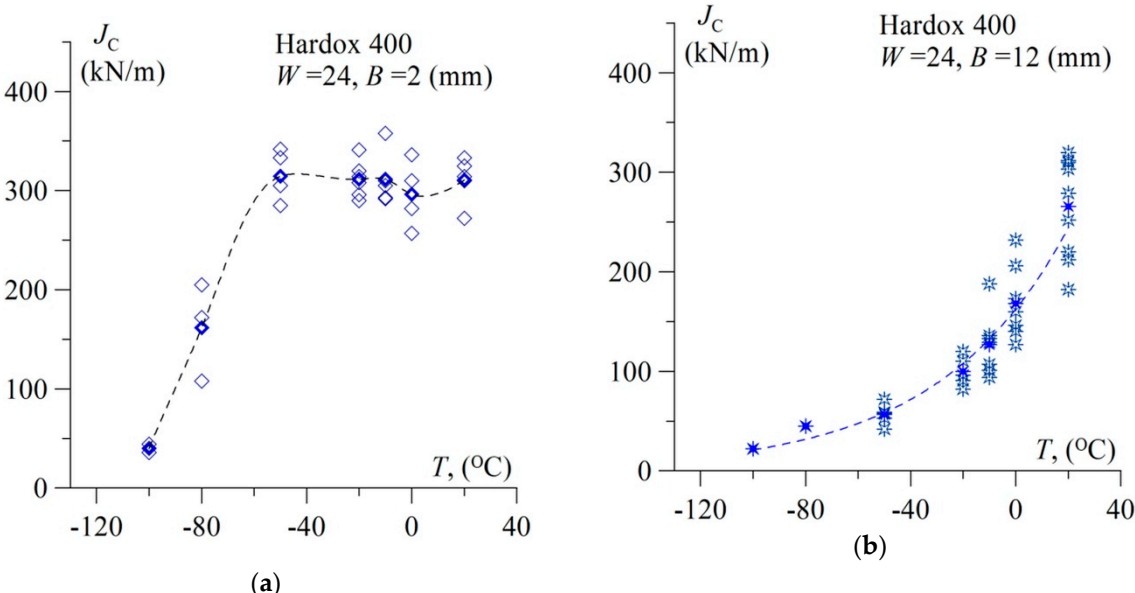

**Figure 5.** The critical values $J_C$ with temperature changes for (**a**) specimens of 2 mm and (**b**) 12 mm thickness.

The distribution graphs of the average critical fracture toughness values $J_C$ of test temperature $T$ in intervals from −100 to 20 °C for different specimens thickness are presented in Figure 6a. In the distribution for specimens of 4 mm thickness, we can observe three regions of the fracture process: Ductile (0–20 °C), ductile-to-brittle (−60–0 °C), and brittle (−100−−80 °C). For thick specimens, which are less than 4 mm-thick, in $J_C$ of $T$ distributions were clearly present in two regions: Ductile fracture (−60–20 °C) and ductile-to-brittle (−100−−80 °C). The fracture toughness data level on higher plateau, where crack growth occurs according to full ductile mechanisms, decreases with reduction of thickness from 4 to 1 mm. The graphs $J_C$ of $T$ distributions for specimens of 8 mm and more were similar. Lower plateau, when the crack growth occurs to the brittle mechanism, and ductile-to-brittle regions were the two regions presented for specimens of thickness from 8 to 24 mm. For specimens of 12 mm thickness and more, the distributions of $J_C = f(T)$ were similar and practically covered.

The distributions graphs $J_C$ of $B$ for tested temperatures are shown in Figure 6b. For the test temperature of 20 °C, ductile fracture mechanism predominated in crack growth for all specimens' thickness. The critical fracture toughness values, $J_C$, for specimens of thickness from 4 to 24 mm were similar, and equal to about 340–360 kN/m. Only for thick specimens (1 mm and 2 mm), fracture toughness decreases with thickness. The similar critical fracture toughness data, $J_C$, were obtained for all specimens tested at $T = -100$ °C (about 33 kN/m). These data belong to lower plateau, where mechanism of crack extension is brittle. For specimens tested at temperature intervals from −80 to 0 °C, the critical values of fracture toughness changed with thickness. For specimens, which thickness corresponds to standard requirements, obtained data were of the similar level. If the thickness of specimens was a little less than critical standard thickness, the fracture toughness level increased. But for thick specimens' critical values of fracture toughness rapidly decreased. This tendency was most clearly observed for $T = 0$ °C and disappeared with test temperature lowering.

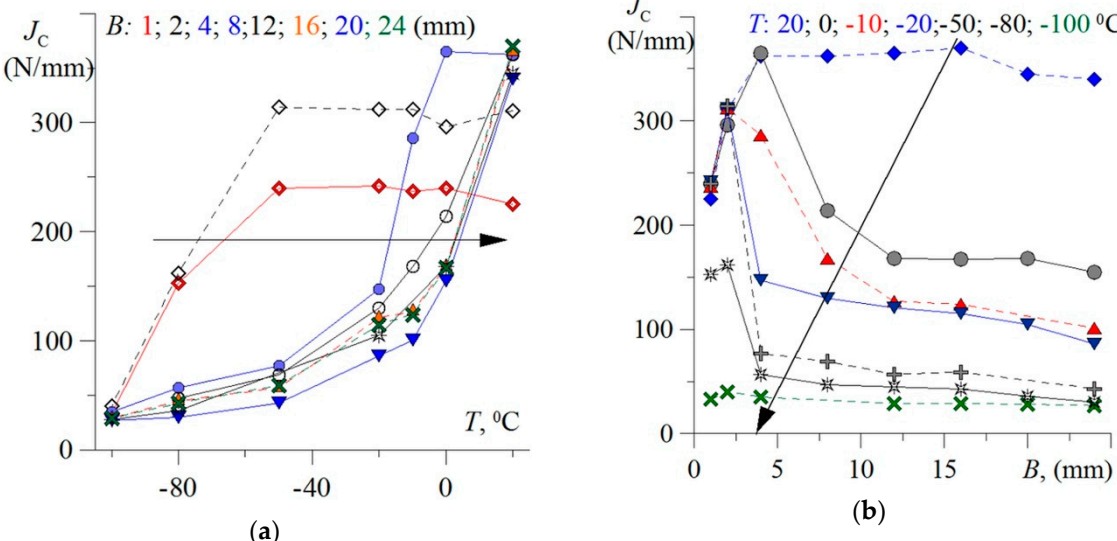

**Figure 6.** (**a**) The $J_C$ with test temperature dependence for specimens of different thicknesses; (**b**) the $J_C$ changes with specimens thickness for different test temperatures.

The differences in morphology of fracture surfaces of specimens of different thicknesses and tested at different temperatures are shown in Figure 7. For all thickness specimens tested at temperatures corresponding to the higher plateau on the fracture surfaces should show the full ductile fracture mechanism of cracks propagations. After wide zone of stretching crack growth by voids mechanisms (voids nucleation on particles of inclusions and precipitates), they rise and coalesce (see Figure 7a,b). At the region of ductile-to-brittle fracture, a stretch zone formed and thin zone with voids were created, then, crack propagated according to brittle mechanism by cleavage (see Figure 7c,d). On the lower plateau cracks grew by fully brittle cleavage cracking (see Figure 7e,f).

The width of stretch zones on fracture surfaces was used to determine the fracture toughness value at the crack initiation moment. From images of the fracture surfaces, it can be shown that a maximum of stretch zone width (*SZW*) corresponded to specimens, that fracture toughness concerns the higher plateau. In the transition region, the width of stretch zone reduced, as shown on fracture surfaces in Figure 7c,d; and when fracture toughness corresponded to lower plateau the *SZW* was inexpressive (weak) (see Figure 7e,f). The details of fracture toughness calculations in the crack initiation moment are presented further in the paper.

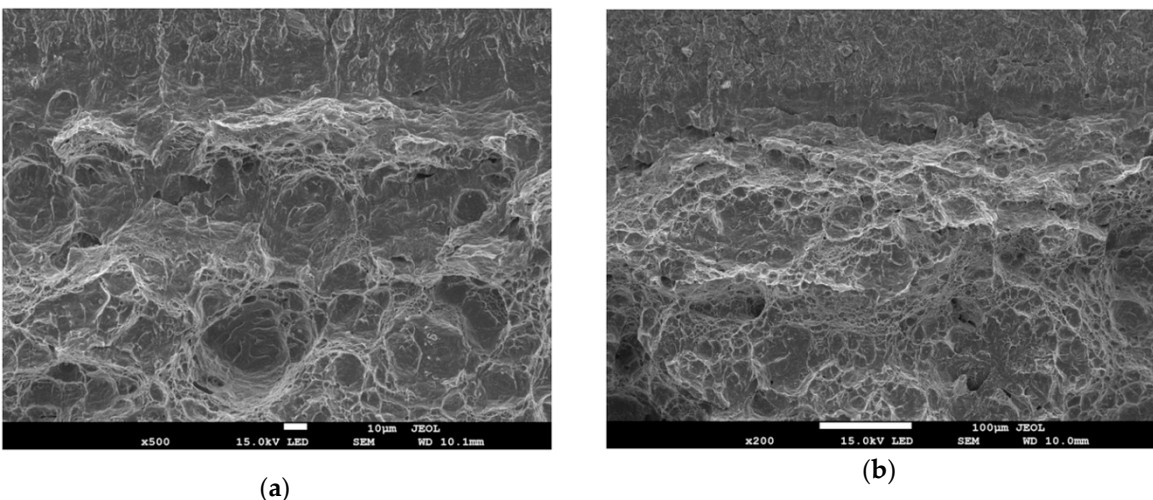

**(a)**    **(b)**

**Figure 7.** *Cont.*

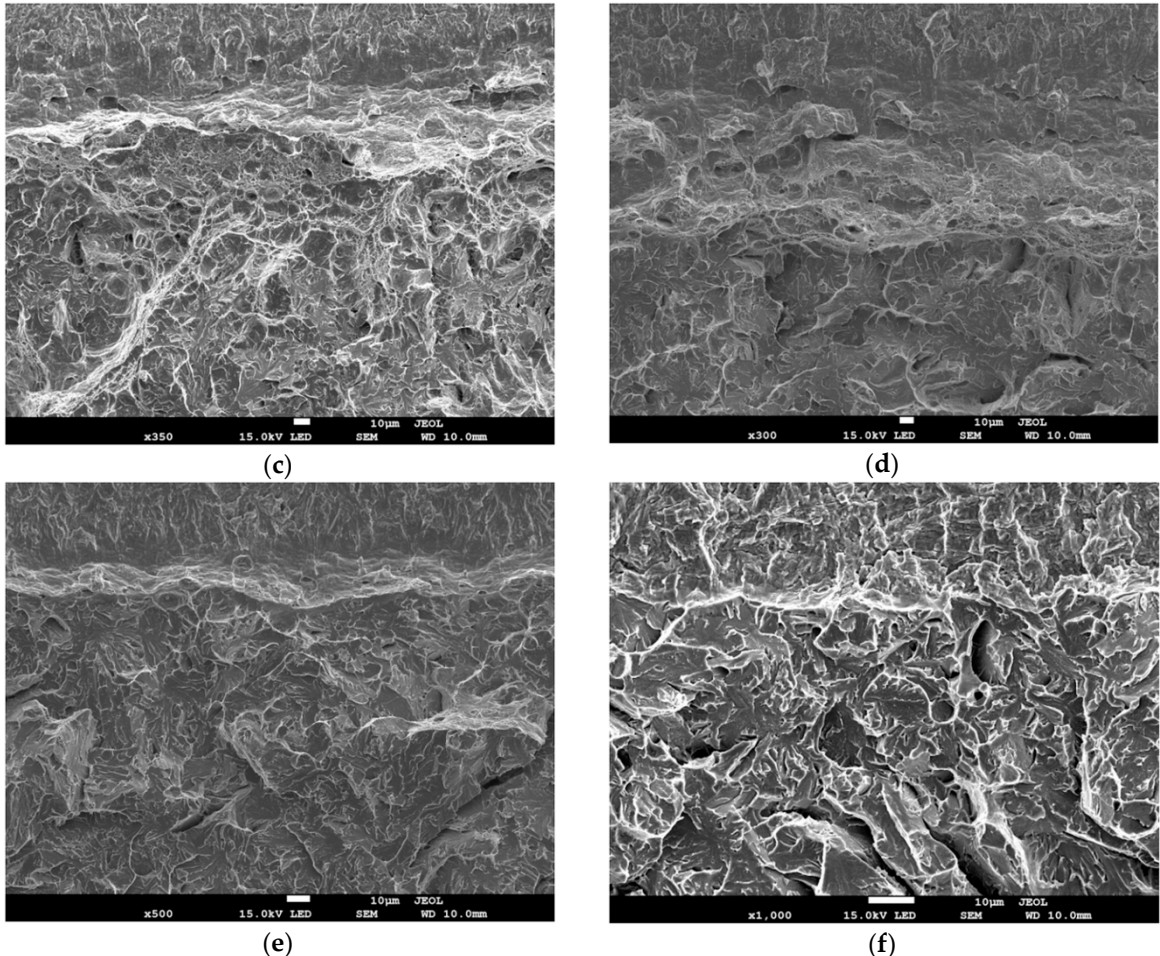

**Figure 7.** Microstructure of Hardox-400 steel (**a**–**c**) with inclusions and (**d**) particles of carbides precipitates in ferritic regions. The fracture surfaces of the specimens: From upper plateau (**a**) $B = 4$ mm, $T_{\text{test}} = 0$ °C and (**b**) $B = 12$ mm, $T_{\text{test}} = 20$ °C; from ductile-to-brittle region (**c**) $B = 8$ mm, $T_{\text{test}} = -50$ °C and (**d**) $B = 12$ mm, $T_{\text{test}} = -20$ °C; from lower plateau (**e**) $B = 2$ mm and (**f**) $B = 12$ mm, $T_{\text{test}} = -100$ °C.

For ferritic steels, the dependence of fracture toughness from temperature in brittle-to-ductile range is recommended to be describe by master curve (MC) [26]. The MC creation procedure is derived from the statistical analysis based on the Weibull model [27–31]. The application of MC is restricted to ferritic steels with a yield strength range between 275 MPa and 825 MPa. The values of yield strength of high-strength steels are higher. The attempts to describe the dependence of fracture toughness from temperature of Hardox-400 steel according to procedures recommended for master curves [26] were not successful. For high-strength steels, such as Hardox-400 and S960QC, other values of transition temperature and minimum value of fracture toughness should be introduced into the formula of master curve. Details of the investigations are presented in the papers [32,33].

*2.4. Local Stress and Strain Distributions in Front of the Cracks*

In the local criteria of fracture analysis, the local distributions of stress and strain before crack tip plays a great importance [34–38]. Based on these distributions, some values should be calculated as stress state triaxiality factors or Lode factor, which make it possible to provide the type of fracture mechanism that is realized during crack propagation. Knowledge of these distributions is required to establish the level of critical stress causing brittle fracture.

The stress and strain distributions before crack tip were calculated using the FEM. Material characteristics were defined based on the true $\sigma$–$\varepsilon$ dependences obtained from a uniaxial tensile test. As a result of numerical calculations, stress distributions in front of crack were determined. The calculations were taken from the model of a three point bend specimen SENB, that has been used during fracture toughness tests (see Figure 8a). Calculations were performed in ADINA program. Length of the fatigue crack was equal to its average value calculated on the basis of standards [24,25]. The crack front was reflected as an arc of 0.01 mm radius (see Figure 8b). The specimen in thickness direction was divided into 11 layers, and distances between the layers decreased in the surface direction. The ratio of the thickness of the outer layer and the middle layer of the sample was 0.3. Size of the mesh elements decreased with approaching to the crack front. The zone surrounding the crack tip with a radius of 2 mm was divided in a radial direction into 30 sections with a density towards the tip of 14. In the model were applied 8-nodes, three-dimensional finite elements. In a definition of boundary conditions was a blocked possibility of a displacement of the specimen surface that is common with the crack front z, the surface in the specimen axis from x direction, and the supporting roll was completely immobilized. In the calculations was steered the displacement of the roll that loaded the tested specimen. The value of displacement corresponded to the initiation moment of the subcritical crack, determined as a result of the experimental research. The model of a large strain was adopted. The true $\sigma$–$\varepsilon$ graphs were used as a material model for the numerical calculation.

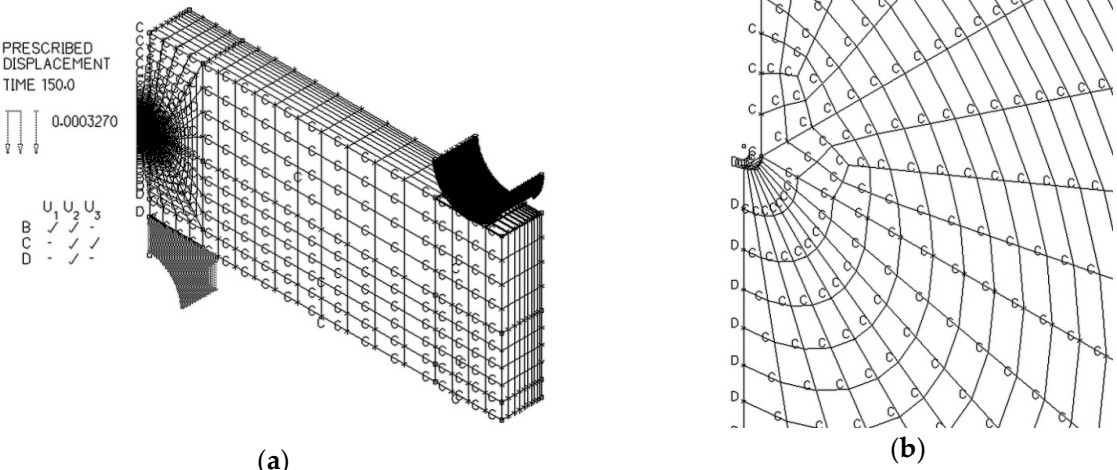

**Figure 8.** (**a**) The SENB specimen model and mesh used in numerical calculation; (**b**) crack tip zone details.

## 3. Results

The local stress and strain distributions before crack front in the critical moment of subcrack initiation are shown on graphs in Figure 9. Stress distributions for three perpendicular directions with distance from crack tip $r$ at $T_{\text{test.}} = -100\ °C$ in the middle plane of specimen are presented in Figure 9a (where, $\sigma_{11}$ is a stress in plane of crack in accordance with growth direction, $\sigma_{22}$ is a stress perpendicular to crack plane in direction which opening crack, and $\sigma_{33}$ is a stress in plane of crack in direction of thickness). The stress distributions at other test temperatures are qualitatively similar. The point of maximum stress distributions moves from crack tip with test temperature increase, as shown for the opening stress $\sigma_{22}$ in Figure 9b. The opening stress $\sigma_{22}$ is the highest and most important in the fracture analysis. In the next analysis, attention will be concentrated mainly on the opening stress $\sigma_{22}$.

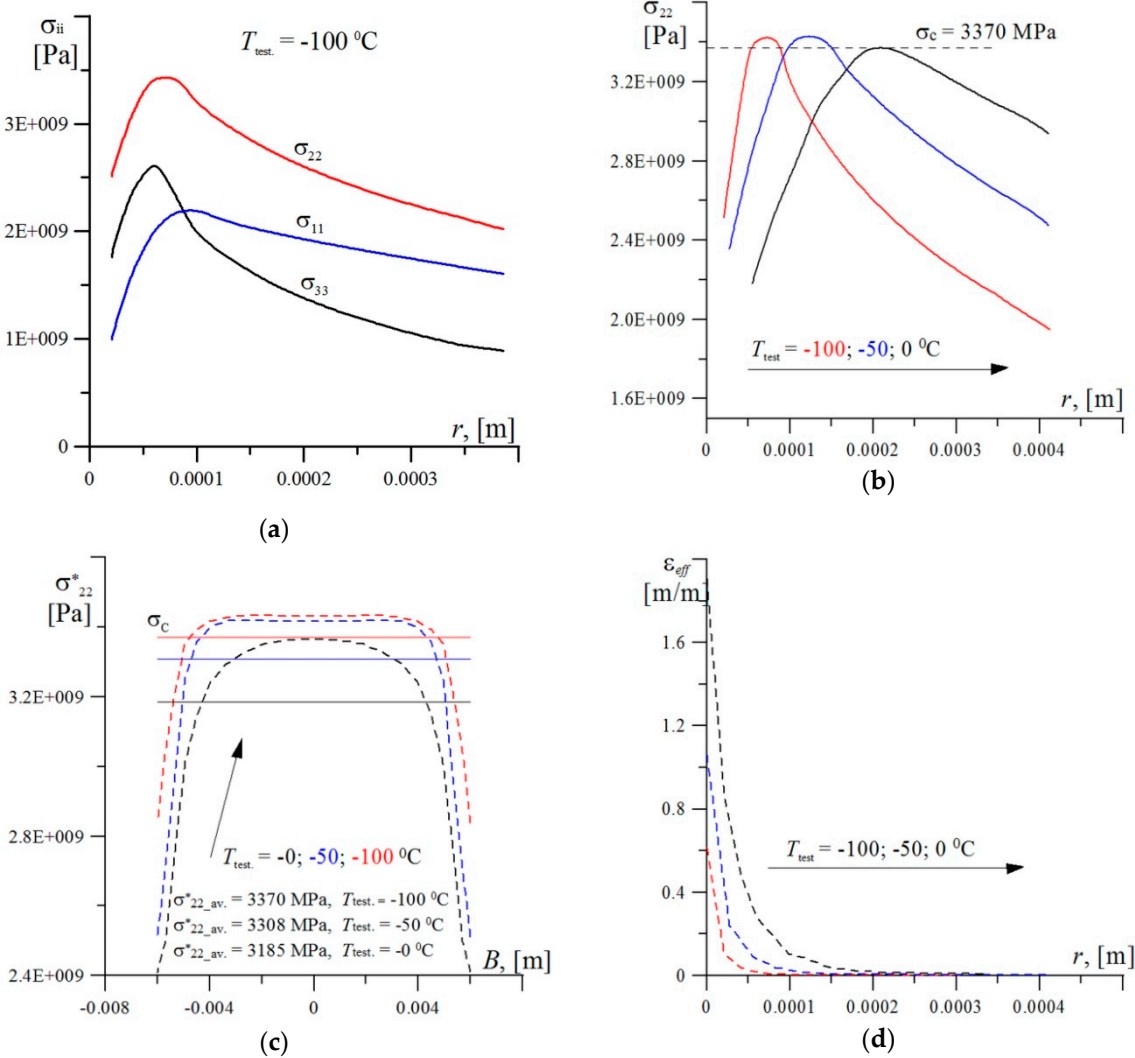

**Figure 9.** (**a**) The distributions of stress components before crack tip at $T_{test.} = -100$ °C; (**b**) the distributions of the opening stress $\sigma_{22}$ at different test temperature; (**c**) the distributions of max values $\sigma^*_{22}$ through thickness; (**d**) the strain distributions at test temperatures.

## 4. Discussion

According to the model of a brittle fracture [34,37,38], it could happen if opening stress $\sigma_{22}$ distribution exceeds the critical level $\sigma_C$ on critical distance $l_C$. Full brittle fracture was observed during specimens tested at $T_{test.} = -100$ °C (see Figure 7e,f). At this $T_{test.}$, the maximum value of the opening stress reached the level of more than $\sigma^*_{22} = 3440$ MPa (see Figure 9a). The distributions of maximum opening stress $\sigma^*_{22}$ values before crack front through thickness of the tested specimens are presented in Figure 9c. Taking into account the fact that at $T_{test.} = -100$ °C, fracture of tested specimens was fully brittle, the average level of $\sigma^*_{22}$ through thickness can be considered as critical stress level, so that $\sigma_C$ needs to realise brittle fracture. For Hardox-400 steel the critical stress level is determined as $\sigma_C = 3370$ MPa. The average levels of $\sigma^*_{22\_av.}$ through thickness distributions for different $T_{test.}$ shown in Figure 9c.

The critical stress level $\sigma_C$ as straight line is presented in Figure 9b,c. The stress distributions $\sigma_{22}$ and $\sigma^*_{22}$ for specimens tested at $T_{test.} = -100$ °C and $-50$ °C exceed the critical stress $\sigma_C$ on some regions, which are conditions for the realization of brittle fracture. While the stress distributions $\sigma_{22}$ and $\sigma^*_{22}$ for specimens tested at $T_{test.} = -0$ °C puts below the critical stress level $\sigma_C$, thus the fully ductile fracture mechanism must be realized during crack growth. The same ductile mechanism of

fracture by growth and coalescence of voids had taken place for specimens tested at $T_{\text{test.}} = -0\,^{\circ}\text{C}$ and $20\,^{\circ}\text{C}$ (see Figure 9a,b).

The effective strain distributions presented in Figure 9d. Comparing the stress and strain distributions (see Figure 9b,d), we can observe that strain reached high values immediately before a crack tip, where stress was significantly decreased. In segments where the opening stress $\sigma_{22}$ exceeded the critical level $\sigma_C$, the strain was low, less than 0.03 m/m. So, for the realization of mechanism by brittle fracture, there must be present high level of opening stress and insignificant strain level. Under the test temperature conditions $-100\,^{\circ}\text{C}$, the low strain level of 0.4–0.6 m/m presented before crack tip and specimen breakthrough was characterized by nonsignificant plastic deformations of the stretch zone width about 5–15 µm (see Figure 7e,f).

With the rise of test temperature, the strain level before crack tip increased also and lead to more developed fracture by ductile mechanisms. On the photograph in Figure 7c, the width of stretch zone is equal about 20 µm and it was formed at strain level of 0.4–1.0 m/m for specimen tested at $T_{\text{test.}} = -50\,^{\circ}\text{C}$. On the fracture surface of specimen tested at $T_{\text{test.}} = -20\,^{\circ}\text{C}$, there can be observed a stretch zone of 20–25 µm width and fragments of the area of crack fracture surface by mechanisms of growth and coalescence of voids (see Figure 7c,d). For specimens tested at $T_{\text{test.}} = 0\,^{\circ}\text{C}$, the *SZW* formed at strain level from range 0.40–1.8 m/m and it was equal about 40–45 µm (see Figure 7a).

Influence of specimen thickness on stress and strain distributions in the area before crack tip was also carried out. The results of calculation for the selected specimens of different thickness and tested at various temperature are presented in Figure 10.

According to numerical calculation results, brittle-by-cleavage fracture of all tested thickness can take place only at $T_{\text{test}} = -100\,^{\circ}\text{C}$ or lower. In this case, the opening stress $\sigma_{22}$ exceeded the critical level $\sigma_C$ for specimens of all thickness (see Figure 10a,d). Investigation of fracture surfaces confirm this results (see Figure 7e,f). At $T_{\text{test.}} = -80\,^{\circ}\text{C}$ in specimen of $B = 2$ mm sharply dropped the $\sigma_{22}$ stresses to the level below $\sigma_C$ (see Figure 10b,d), while effective strain $\varepsilon_{\text{eff}}$ rapidly grew (see Figure 10e)—this means that ductile mechanism of subcrack propagation will be implemented. This suggestion is confirmed by the experimental results shown in Figure 6.

Temperature increase caused a reduction of the level of $\sigma_{22}$ stresses and replacement of fracture mechanism from brittle on ductile, especially in thinner specimen. According to the data presented in Figure 10c, at test temperature $-20\,^{\circ}\text{C}$ in specimens of thickness less than 4 mm should occur ductile fracture mechanisms of subcrack propagation. In specimens of thickness, more than 4 mm brittle fracture may occur. While in the area between crack tip and region, where $\sigma_{22} > \sigma_C$, current high level of strains (see Figure 10f), the two types of fracture will be present in crack propagation—ductile and brittle (see Figure 7c,d).

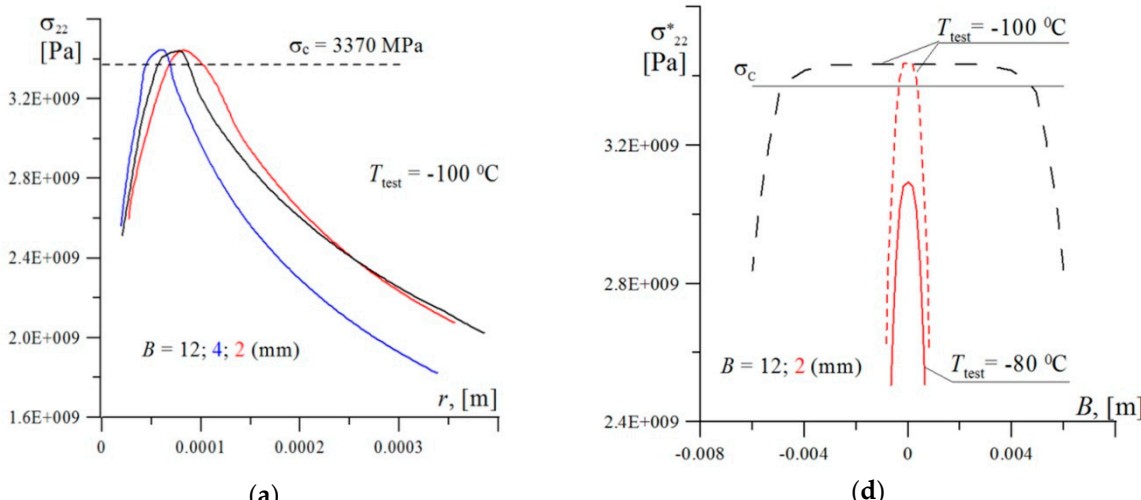

**(a)**　　　　　　　　　　　　　　　　　　　　　　　　**(d)**

**Figure 10.** *Cont.*

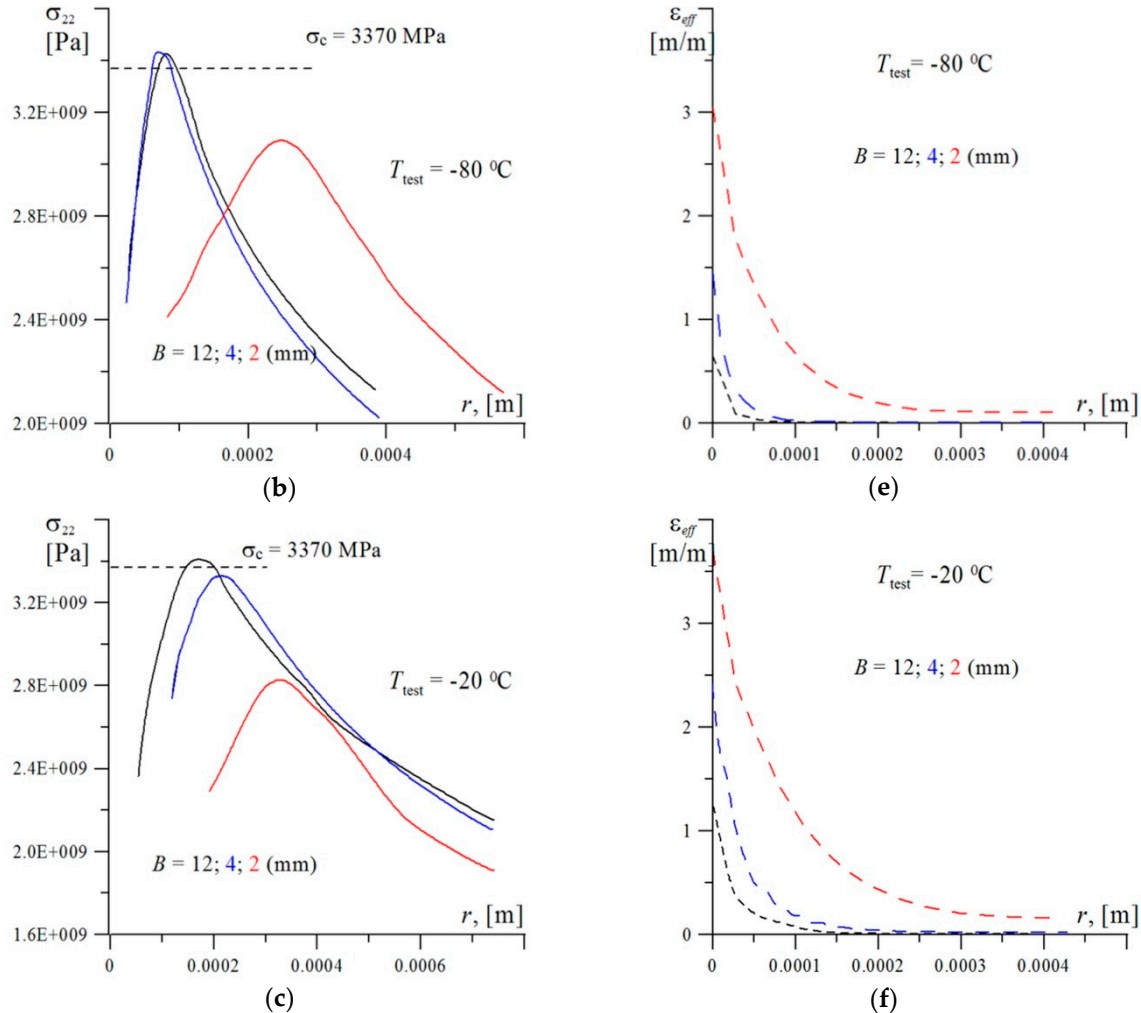

**Figure 10.** (**a**) The distributions of $\sigma_{22}$ for specimens of $B = 12.\ 0$, 4.0, 2.0 (mm) at $T_{\text{test}} = -100\ °C$; (**b**) $-80\ °C$; (**c**) $-20\ °C$. (**d**) The distributions of $\sigma^{*}_{22}$ through thickness for specimens of $B = 12.\ 0$ and 2.0 (mm) at $T_{\text{test}} = -100\ °C$ and $-80\ °C$. (**e**) The distributions of $\varepsilon_{\text{eff}}$ for specimens of $B = 12.\ 0$, 4.0, 2.0 (mm) at $T_{\text{test}} = -80\ °C$ and (**f**) $-20\ °C$.

The procedure for the determination of the critical fracture toughness value at the moment of start of subcrack, $J_i$, is based on the determination of the stretch zone width ($\Delta a_{\text{SZW}}$), which occurs prior to the subcritical crack initiation [39–41]. When a subcrack initiation is preceded by significant plastic strain, the formula proposed by Shih [42] is the most appropriate one:

$$J_i = (2\sigma_f/d_n)\Delta\bar{a}_{\text{SZW}},\tag{1}$$

where, $\sigma_f = 0.5(\sigma_y + \sigma_{ut})$; $d_n$ is a function dependent on the material hardening coefficient $n$ and triaxiality coefficient $T_Z = \sigma_{33}/(\sigma_{11} + \sigma_{22})$ [43]. Values $T_z$ were computed based on stress distribution data, which were determined in numerical methods. Coefficient $d_n$ determined on the basis of formulas proposed by Guo [44]:

$$d_n = d_0 - (2T_z)^{a_3(n)}(d_0 - d_5),\tag{2a}$$

for $n > 5$:

$$
\begin{aligned}
d_0 &= 1 - 0.1240(1/n)^{1/2} + 0.8968(1/n) - 13.3941(1/n)^{3/2} + 15.3139(1/n)^2;\\
d_5 &= 0.78 + 0.0277(1/n)^{1/2} - 3.0791(1/n) + 2.4709(1/n)^{3/2};\\
a_3(n) &= 11.4 - 45.8(1/n)
\end{aligned}\tag{2b}
$$

The view of stretch zone obtained by SEM is shown in the upper part of the photo in Figure 11a. At the critical moment, the microcracks or voids of the damaged material before the blunted crack connects with the blunted crack and subcrack propagation starts. Based on the measurement of *SZW*, the critical values of fracture toughness at crack initiation, $J_i$, were estimated. The details of *SZW* measurement are presented in the papers [45,46]. The critical values of *J*-integral at the subcrack initiation, $J_i$, in the temperature range for which the growth proceeds according to the mixed mechanism (−20–20 °C), are located lower than $J_{IC}$ (see Figure 11b). The difference increases with a rise of the test temperature. The reason why differences in $J_{IC}$ and $J_i$ values occur is the fact that for ductile growth of the subcrack, the values of $J_{IC}$, are characterized of fracture toughness for averaged extension of $\Delta a$ = 0.2 mm length, while $J_i$—at the instant of initiation.

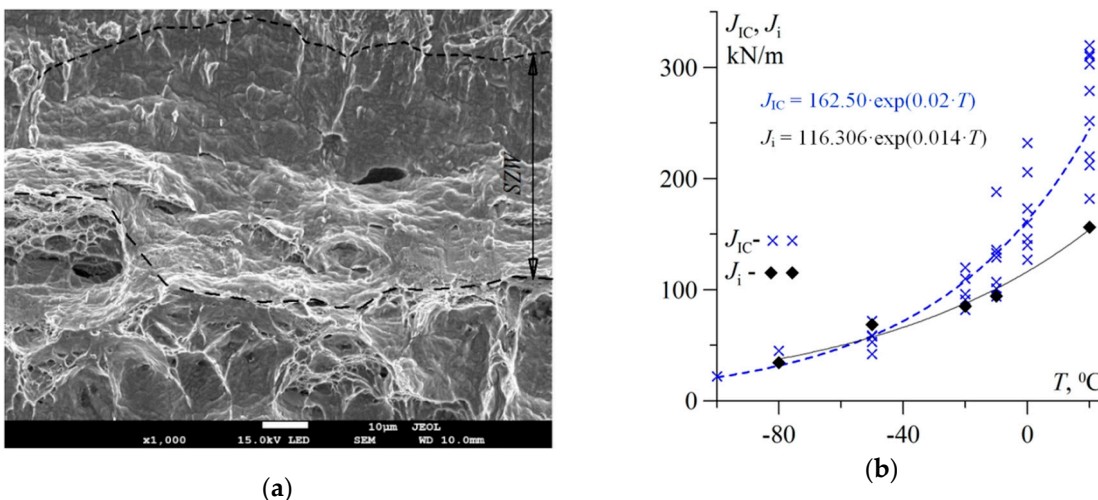

(**a**)     (**b**)

**Figure 11.** (**a**) A stretch zone view; (**b**) a comparison of the critical values of the $J_{IC}$ and $J_i$.

## 5. Conclusions

As a result of the conducted research, dependencies of strength characteristics and fracture toughness in the range of operating temperatures (−100–20 °C) of Hardox-400 steels were determined. Also in this temperature range, the effect of thickness on the critical value of fracture toughness was investigated. The obtained results will allow to perform strength analysis of structural elements made of Hardox-400 steel according to FITNET [22] or other similar procedures. The results obtained during the examination of mechanical properties of Hardox-400 steels allowed to formulate several detailed conclusions presented below.

Hardox-400 steel is low-carbon steel which was produced using a controlled thermomechanical treatment that leads to formation of microstructure of tempered bainite–martensite, which contains numerous particles of carbides precipitates and separate isolated large nonmetallic inclusions (see Figure 1). This steel is characterized by high level of strength properties and relatively good level of plasticity (Table 2). Attention should be paid to the fact of increasing the plasticity with lowering test temperature. Also it is important to note that the hardness in the middle layers of the plate is significantly reduced. Lowering hardness of steel leads to a slight reduction of strength properties and slightly larger changes of fracture toughness.

The test temperature and thickness of specimens have a significant influence on the level of fracture toughness characteristics of Hardox-400 steel (see Figure 5). But this influence is not stable. High-strength steel of similar microstructure (S960QL) tested at room temperature was characterized of ductile fracture mechanism despite of the occurrence of the local areas of brittle fracture [47]. Generally, with test temperature decrease, fracture toughness characteristics of Hardox-400 steel decreases too. Such trends correspond to specimens, in which plane strain is dominant and the condition on the specimen thickness, $B \geq 25 J_{IC}/\sigma_y$, is fulfilled. But for specimens, in which plane strain is not dominant,

the critical fracture toughness values show similar high values of $J_C$ and located on higher plateau of brittle-to-ductile dependence. When more restrictive conditions on the specimen thickness were fulfilled, $B \geq 2.5(K_{IC}/\sigma_y)^2$, the fracture toughness values regardless of specimens' thickness were similar and located on the lower plateau. In this case, a full brittle-by-cleavage mechanism fracture takes place in tested specimens.

Brittle fracture is very dangerous, because it causes immediate destruction of elements. In order to assess the occurrence of a critical situation in which brittle fracture takes place, numerical FEM modelling and calculation of local stress distributions before the crack tip were performed. Based on the analysis of stress distributions and fractographic tests of specimens' fracture surfaces, the critical level of opening stress $\sigma_{22}$ for Hardox-400 steel was determined as $\sigma_C = 3370$ MPa. If the stress $\sigma_{22}$ exceeds the critical level $\sigma_C$, a brittle fracture occurs in the appropriate area and it grows in net cross section of the test specimens. When the stresses $\sigma_{22}$ were lower of the critical level $\sigma_C$, a ductile fracture mechanism of crack growths by voids nucleation, growth and coalescence were observed. Hardox-400 steel contains large particles of inclusions, which additionally increases the level of local stresses. This is the reason why, in steel, there is a large data scatter of critical fracture toughness values $J_C$ at test temperatures higher than $-50$ °C.

Joint consideration of stress and strain levels in the specimens allows us to determine the subcrack growth mechanism: Full brittle, brittle-ductile, and full ductile. The results of numerical analysis were confirmed by the results of experimental research and the observation of fracture surfaces of specimens on the SEM.

**Author Contributions:** Conceptualized research, methodology and wrote article, I.D.; writing-review & editing, investigation, I.D. and R.P.; data curation, visualization R.P.

**Funding:** The research was financed by the National Science Centre, Poland (No. 2017/25/N/ST8/00179).

**Acknowledgments:** Authors would like to thank P. Furmanczyk for help in carrying out research by SEM.

**Conflicts of Interest:** The authors declare no conflicts of interest.

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
