# Peer review of "Strength and Fracture Toughness of Hardox-400 Steel"

_metals, doi:10.3390/met9050508_

Reviewer 1 Report

Dear Authors,

This work can be labelled as an experimental rather than scientific paper the original contribution of which is limited.

Following you can find some commends improving the quality for your work.

1)     The introduction is too short – it will be more valuable to perform a greater literature review over pointing the “state of the art” in this matter.

2)     In line 50 it is written that the particels of titanium nitrides enriched with Nb, How could you conclude it? Did you perform the EDX Experiments?

3)     In order to make figure 3 understandable in Black-white version, please change the format of the lines (e.g. dash line, …)

4)     There is a lack of information about the fracture toughness measurment, what ist the specification of the testing equipment?

Author Response

Thank You very much for your review and comments of our paper.

1.The Introduction has been rewritten according to your suggestions. Introduced into the literature review relationship between microstructure and properties of high-strength steels.

2.The results of EDX analysis added to explaining content of inclusions.

3. Presenting results by lines in color is accepted of Metals journal.

4. Some information according to test equipment is introduced into text. All experimental tests carried out according to proper ASTM Standards.

5. English language and style of paper are corrected additionally.

Reviewer 2 Report

The topic is not highly innovative, but it aims to complements the knowledge about a particular steel grade. Numerical FEM tests setup lacks details, such as mesh density, finite element type and material law, so that should be complemented for the final version.

Also, authors could be more specific stating where this study particularly exceeds other authors concerning Hardox steel charahterization.

Author Response

Thank You very much for your review and comments of our paper.

1. Introduced into text some details according to FEM tests, as mesh density, finite element type. The true stress-strain relationships for according temperature were used as material law in FEM.

2. Results of fracture toughness tests of high strength steels are rarely presents, especially for wide temperature range and different plate thickness. For Hardox-400 too. Authors presented results for this grade steel, which give complete data and could be used for assessment of strength different construction elements made of this steel.

3. English language and style of manuscript are corrected additionally.

Reviewer 3 Report

The authors report an interesting study on the Strength and fracture toughness of HARDOX-400 steel. The study has been carried out using the state-of-the-art techniques. The reported conclusions are sound and well supported by the reported evidence. My opinion is that the article can be published. However, several minor changes are needed to improve it.

1.       In the introduction, at the beginning of the first paragraph a sentence on the importance of the study of the mechanical properties on metallic alloys for the aerospace and automotive industry. This should be supported by references: A. Grajcar et al. Metals 2018, 8, 1028; D. Smith et al. J Phys Condens Matter. 2017, 29, 155401; S. Djebali et al. Procedia Engineering 2015, 114, 306. By adding this sentence, the article will have a more general approach, reaching a broader audience.

2.       Figure 1 should be rearranged to fit in only one page.

3.       Provide the error for magnitudes given in Table 2.

4.       Provide the error on temperature in different figures. If smaller than symbols size please state it.

5.       Figure 6 should fit in only one page. Same for Figure 9.

6.       In Fig. 10 Jic should be 162.50 x exp(0.02xT). The same for Ji. The present form of the equations could lead to confusion.

Author Response

Thank You very much for your review and comments of our paper.

1. The Introduction is written according to your suggestions. Added in Introduction review of literature some references according to microstructure-properties relationships of high strength steels.

2. The minor corrections of manuscript, as placing figures in one page, add the scatter of results and temperature error, are introduced.

3. The equations in fig. 11b is present in form, which was suggested of Reviewer.  

4. Some corrects of English language and style of paper made additionally.